# How has COVID-19 changed healthcare professionals' attitudes to self-care? A mixed methods research study

Peter Samuel Smith[1,2]*, Aos Alaa[2], Eva Riboli Sasco[2], Emmanouil Bagkeris[3], Austen El-Osta[2]

**1** The Self Care Forum, Surrey, United Kingdom, **2** Self-Care Academic Research Unit (SCARU), Department of Primary Care & Public Health, School of Public Health, Imperial College London, London, United Kingdom, **3** National Heart and Lung Institute, Imperial College London, London, United Kingdom

* p.smith@imperial.ac.uk

## Abstract

### Background

The COVID-19 pandemic fundamentally changed the way services are delivered. Self-care, including good hygiene practices and avoidance of risk was emphasised as the key measure to tackle the pandemic in the early stages.

### Objective

To understand how self-reported professional attitudes, perceptions and practices of self-care have changed as a result of the COVID-19 pandemic.

### Design

Cross-sectional online survey and semi-structured qualitative interview.

### Setting

Health care.

### Participants

304 healthcare professionals (HCPs).

### Methods

A wide range of HCPs, including pharmacists, nurses, doctors, social prescribers and other designations took part in a 27-item anonymous online survey. Semi-structured qualitative interviews with nine healthcare professionals explored attitudes to and practices of self-care before and during the pandemic. Views were sought on the permanence and implications of changes. Data were analysed using routine statistics and thematic analysis to identify major themes.

**Data Availability Statement:** The data underlying the results presented in the study are available in Supplementary File S6 File. RAW_DATA.

**Funding:** The author(s) received no specific funding for this work.

**Competing interests:** The authors have declared that no competing interests exist.

## Results

A total of 304 HCPs responded to the survey fully. Nine participated in a semi-structured interview. There was agreement that the importance of self-care has increased markedly during the pandemic. The percentage of respondents who felt that self-care was 'very' important to their clients increased from 54.3% to 86.6% since the pandemic. Personal empowerment and capacity of service users to self-care increased significantly during the pandemic. Willingness of patients to engage (74%) and poor understanding of self-care (71%) were cited as the two main barriers to self-care. A close third was digital exclusion (71%), though 86% of respondents recommended online resources and 77% the use of smartphone apps. Survey respondents believed the changes to be permanent and positive. Interviewees reported a major, and positive move to self-care with the pandemic seen as an opportunity to be grasped, but professional education would have to be aligned to make the most of it. They raised concerns as to whether the shift to self-care was perceived by users as 'abandonment' rather than 'empowerment' and whether problems had been stored rather than dealt with through self-care and therefore whether the positive changes would continue after the pandemic.

## Conclusion

Reporting their views before the pandemic, barely more than half of the professionals surveyed saw self-care as fundamentally important to the individuals they served. This changed to 86% as a result of the pandemic. Patient/client engagement with and understanding of self-care were reported as major barriers, as was digital exclusion, though increased technological solutions were used by all respondents. Concerns were raised that the permanence of the changes depended upon continued encouragement and empowerment of individuals to self-care and on its inclusion in professional education as a substantive subject.

## Introduction

The COVID-19 pandemic fundamentally changed the way health, care and wellbeing services were accessed and delivered in many parts of the world [1–3]. Many services rapidly changed to remote delivery models following the reduction or temporary abandonment of face-to-face and hands-on care [4]. For the vast majority of the UK population, and internationally, self-care rapidly became a mainstay of service delivery, especially during the national lockdowns to reduce the spread of SARS-CoV-2 [5].

Initially, professionals and the public alike appeared largely unequipped to handle the new dynamic for immediate and ongoing care. Within weeks of the World Health Organisation (WHO) declaring the pandemic, changes in policy and professional attitudes highlighted the singular importance of self-care as a first choice in tackling COVID-19 as individuals and society were supported to self-test using rapid antigen lateral flow test (LFT) kits, recommended to wash hands regularly and wear personal protective equipment. Increasing awareness, coupled to the rational use of products and services (including self-testing using LFT for example), risk avoidance and good hygiene practices are in fact key pillars of self-care [6], and were the only line of defence against the communicable disease until safe and effective vaccines, prophylaxis and pharmacological treatments became available.

Responses to an invitation to apply for The Self-Care Forum Coronavirus Innovation Award in 2020 [7] during the first UK national lockdown suggested there had been a rapid and creative response from service providers to incorporate self-care into mainstream delivery. Concomitantly, numerous changes were made to delivery; most of the standard written advice provided to clinicians working for NHS 111 to convey in-calls to the Coronavirus Clinical Assessment Service were specifically delivered through 'self-care' advice [8, 9] at a time when general practitioners (GPs) and physiotherapists moved largely to online access [4, 10, 11].

Evidence from the Innovation Awards suggests that HCPs adapted their expectations and practices to support new ways of working [7], but little is known about this quantum change in professional attitudes and practice, its likely impact and whether any of the changes that manifested as a result of the healthcare emergency will persist post-pandemic.

A paper by Godfrey et al. highlighted 139 extant definitions of self-care in the academic literature in 2011 [12]. Even the WHO has 5 definitions of self-care (the latest being proposed in 2019). The current WHO working definition of self-care is "the ability of individuals, families and communities to promote their own health, prevent disease, maintain health, and to cope with illness and disability with or without the support of a health worker" [13].

There is ample literature supporting the argument that self-care can improve the health and lives of individuals whilst helping reduce pressure on scarce NHS resources [14, 15]. Surveys of the public on the need to self-care as a result of the pandemic reported significant changes in personal attitudes [2, 3, 16]. Although this seems a step in the right direction, it falls short of the 'fully engaged' scenario described in the Wanless Review of 2002. The review presented a view of the 'fully engaged' self-caring individual as an essential requirement for the survival of a thriving NHS during the 20 years of its projections from 2002 to 2022 [17].

Knowledge about how health and social care professionals feel about self-care is limited. There is scant research that focuses on the attitudes of HCPs in any setting regarding self-care since the first national lockdown and as a result of the COVID-19 pandemic. The aim of this study was to explore how the prevailing attitudes and practices of HCPs regarding self-care may have changed since the advent of the COVID-19 pandemic. Specifically, we sought to understand how confident HCPs feel in recommending self-care & lifestyle interventions in their line of work, and if reliance on self-care has become more common practice. We also investigated HCPs' attitudes to their own self-care and the impact technology has had on their routine interactions with patients.

## Materials and methods

### Study design

The study adopted a mixed methodology, using a cross-sectional online survey and semi-structured interviews from a mix of doctors, nurses, pharmacists, social prescribers (SP) and other professionals from the UK health and social care setting. For the interviews, we selected a sub-sample to incorporate a mix of clinical roles and specialist interest in self-care.

The link to the electronic survey was published and available on the Imperial College Qualtrics platform between 3 February and 3 August 2021 (6 months). The voluntary survey was open and could be accessed by anyone with a link. Potentially eligible participants received an invitation email from the study team, and the Self Care Forum also disseminated the email and link to a network of health and social care professionals. Study information was disseminated, including the Participant Information Sheet (PIS) and link to the survey. The researchers' personal and professional networks were also mobilized to respond and further disseminate the eSurvey among potentially eligible participants. The PIS included information regarding the study's aims, the protection of participants' personal data, their right to withdraw from the

study at any time, which data were stored, where and for how long, who the investigator was and survey length. Participants were informed that this was a voluntary survey without any monetary incentives but offering the possibility to access the findings at a later stage whilst underlying the potential collective benefits of taking part in terms of helping advance knowledge in this area and the formulation of future policies to tackle the COVID-19 pandemic. The data collected were stored on the Imperial College London secure database, and only the team researchers could access the eSurvey results.

All responses were pseudo-anonymised to ensure confidentiality by assigning each respondent a unique study ID. Only the participants' demographic data (age in years, gender, ethnicity, occupation/ designation, and the first segment of postal code) were recorded.

**Electronic survey.** The open survey comprised a total of 27 questions displayed across several screens and was accessible using a personal computer or smartphone. The eSurvey can be accessed in **S1 File** or by following this link:

https://imperial.eu.qualtrics.com/jfe/form/SV_1AL3nq2UJDCWgjH

Questions regarding demographic characteristics of the users included information on gender, age, ethnicity, occupation/field of work and the first part of postal code. Participants could review their answers before submitting them. All data collected through the survey were anonymised and not personally identifiable. All data, including uncompleted surveys, was collected. The online survey technical functionality was tested before being published. The first question asked participants to confirm their consent to participate in the eSurvey.

Changes in the delivery of care, with respect to self-care interventions offered by health and social care staff to their service users as a result of the COVID-19 pandemic were gleaned through several questions concerning self-reported or perceived levels of care, pre-and in-COVID. Respondents were presented with questions including a list of different aspects of self-care from which they could select a number of options. Questions concerning health & social care professionals' attitudes and perceptions towards the effect of the pandemic on their approach to self-care procedures, both offered to service users and in their own personal lives, were scored on a 1–5 Likert scale. Respondents were able to refrain from providing an answer by selecting 'no opinion'. Such answers were treated as missing data in all the analyses (listwise exclusion), but due to the small number of missingness (<1.5%), the data were not imputed [18, 19].

The survey included two questions to explore the perceived barriers towards self-care procedures, including questions regarding technological barriers (e.g. digital exclusion), systemic barriers (e.g. funding) and professional barriers (e.g. interprofessional communication). An additional 1–5 Likert scale question was included regarding the opinion of HCPs' predictions on their attitudes towards their adhesion to self-care procedures post-COVID-19. Respondents were also asked to rate the capacity and empowerment of their service users to self-care from 1–10 on a Likert scale, both before and during the pandemic.

**Personal interviews.** One-to-one interviews were conducted between 28 July and 8 September 2021 by PS (a male general practitioner), remotely through video or telephone. The interviewees were not selected but were recruited as a convenience sample from those who volunteered to partake in the study with a sample of 9 participants meeting the inclusion criteria (over 18 years of age, English-speaking, and currently working in nursing, health, pharmacy or social care setting). Participants who did not consent to be interviewed were excluded. Participants were told who the interviewer was, the purpose of the study, the length of time of the interview, where the data was stored and for how long. They were able to withdraw at any point, without giving a reason leading up to, or during the interview; advised not to answer questions they were uncomfortable with, without giving any explanations. All participants provided consent via Interview Consent Form to the publication of their anonymized responses.

Interviews were digitally recorded and transcribed; transcripts were not repeated or returned to the participants. Participants were unknown to the interviewer prior to the interview. Field notes were used to record relevant contextual issues. All interviews took place ensuring privacy, with no one else present apart from the participant and interviewer. All interviewees completed the interview and answered all the questions without a break or terminating the interview. The interview process was terminated when no new information was forthcoming, and data saturation was reached. The duration of the interviews varied between 15 and 45 minutes, with the average being 35 minutes. The conversation was audio-recorded with consent, and the recording was destroyed after transcription. All data and interview transcripts were stored securely on an encrypted and secure institutional server, which can only be accessed using passwords adhering to Imperial College London policies and procedures. Participants were informed that any ad verbatim quotations that might be included to illustrate key themes would be anonymised. The interview questions were open, semi-structured and designed to explore participant's' experiences, attitudes and views in more depth. The questions were first discussed with the research team (face validity) before drawing up an interview guide which was piloted and refined further based on an in-depth literature review. The guide was developed to provide structure and focus. Probing questions were asked, and participants were encouraged to express additional opinions and comments. A summary of questions asked is included in S3 File. Specifically, our objectives were to: (1) Identify views of HCPs on self-care as a primary mode of care, (2) Explore perceived barriers faced by service users to practice self-care, (3) identify the perceived barriers faced by HCPs, and (4) explore the changes in attitudes towards self-care following the advent of the COVID-19 pandemic.

**Data analysis.** Quantitative data were collected using an eSurvey questionnaire administered on Qualtrics. Survey responses were summarised using frequencies and percentages. Chi-square test was used to compare groups. McNemar's test was used to compare paired data from different time horizons (i.e. before and during the pandemic) regarding HCPs' perceptions of whether self-care was important to service users. A p-value $<0.05$ was considered statistically significant. The factors that were significant in the univariable models (p-value $<0.05$) were considered in the multivariable analyses. All analyses were performed using SPSS (Statistical Package for Social Sciences) version 28.0.1. The Checklist for Reporting Results of Internet E-Surveys (CHERRIES) was used to guide reporting (S5 File) [20].

Contextual data gleaned from personal interviews were analysed according to the principles of interpretive thematic analysis. Interviews were listened to, and transcripts were read several times to capture all the information provided by the participants. Notes were made under each of the four objectives, including ad verbatim quotations using the information provided by each participant. Interview notes were read thoroughly and thematically analysed for their contents (familiarization; generating initial codes; searching for themes; reviewing themes; defining and naming themes). Initially, each transcript was coded using a process of open coding by PS in discussion with AEO, followed by the development and clustering of themes in an interpretive process. The basic codes were elaborated into a framework that was continuously refined to reflect all the interviews (S2 File and S2 Table). The study team did not discuss findings with participants but were keen to share publications with anyone who expressed interest. This method was chosen as it enabled the identification, analysis and interpretation of patterns and meaning within qualitative data. In addition, this design is not tied to a particular epistemological or theoretical perspective making it flexible and appropriate for this study. The emergent themes were checked against the interview guide and study objectives, resulting in the development of a set of major themes. Co-authors of the study verified the emerging themes and contents. The Consolidated Criteria for Reporting Qualitative Research (COREQ) were used to guide reporting (S4 File) [21].

**Ethics.** The study was given ethical approval by Imperial College Research Ethics Committee (ICREC # 20IC6505). Participants consented to take part in the survey.

**Patient and public involvement.** No patient was involved.

## Results

### eSurvey

**Demographic profile of respondents.** The electronic survey captured full responses from 304 respondents who were either medical, nursing, pharmacy or allied HCPs from across England (**Table 1**). Data from a further 20 respondents were excluded from the analysis due to incomplete survey responses or lack of key demographic data, including occupation.

Seventy eight percent of respondents were female, and the majority were white (79.1%). Nearly two thirds (64.2%) worked in the general practice setting, whereas 7.8%, 6.4%, 5.3% and 4.0% worked in secondary care, social care, voluntary sectors and public health services, respectively. Nearly 40% were pharmacy staff where most (81%) were pharmacists. Although doctors only comprised 15.7% of the total sample, the vast majority (88%) were general practitioners. The results of the survey are shown in Table 2 and S1 Table.

**HCP views on the importance of self-care to service users.** The percentage of HCPs reporting that self-care was 'very important' to service users rose significantly from 54.3% pre-pandemic to 86.8% during the pandemic, p<0.001, (Table 3). This is reflected across all professional groups (Table 2). Across all professional groups, 'making healthy lifestyle choices' and

**Table 1. Participant characteristics.**

|  | n | (%) |
|---|---|---|
| **Age** | | |
| 20–29 | 10 | (3.3) |
| 30–39 | 42 | (20.0) |
| 40–49 | 77 | (25.6) |
| 50–59 | 114 | (37.9) |
| 60–69 | 51 | (16.9) |
| 70–79 | 5 | (1.7) |
| 80+ | 2 | (0.7) |
| **Gender** | | |
| Male | 63 | (20.7) |
| Female | 235 | (77.3) |
| Other | 2 | (0.7) |
| Unknown | 4 | (1.3) |
| **Ethnicity** | | |
| Asian/Asian British | 40 | (13.2) |
| Black/African/Caribbean/Black British | 10 | (3.3) |
| Mixed/multiple ethnic groups | 3 | (1.0) |
| White | 240 | (78.9) |
| Other* | 11 | (3.6) |
| **Role** | | |
| Doctor | 47 | (15.7) |
| Nurse | 44 | (14.7) |
| Services & therapy professionals | 41 | (13.7) |
| Pharmacy staff | 119 | (39.7) |
| Social prescriber (SP) or other (i.e., carer, non-GP doctor, commissioner of health) | 49 | (16.3) |

**Table 2.  Results of electronic survey.**

| | Doctor | Nurse | Service | Pharmacy | SP/Other | Total | p-value |
|---|---|---|---|---|---|---|---|
| | N (%) | N (%) | N (%) | N (%) | N (%) | N (%) | |
| **Perceived importance of self-care to patients before the pandemic** | | | | | | | 0.07 |
| Very unimportant | 1 (2.1) | 5 (11.4) | 2 (4.8) | 8 (6.7) | 6 (11.8) | 22 (7.2) | |
| Unimportant | 0 (0.0) | 0 (0.0) | 0 (0.0) | 1 (0.8) | 0 (0.0) | 1 (0.3) | |
| Neutral | 1 (2.1) | 2 (4.5) | 1 (2.4) | 9 (7.6) | 7 (13.7) | 20 (6.6) | |
| Important | 13 (27.1) | 17 (38.6) | 8 (19.0) | 45 (37.8) | 13 (25.5) | 96 (31.6) | |
| Very important | 33 (68.8) | 20 (45.5) | 31 (73.8) | 56 (47.1) | 25 (49.0) | 165 (54.3) | |
| TOTAL | 48 (100.0) | 44 (100.0) | 42 (100.0) | 119 (100.0) | 51 (100.0) | 304 (100.0) | |
| **Perceived capacity and empowerment of patients to self-care before the pandemic** | | | | | | | 0.19 |
| Very unimportant (0–1 of Likert scale) | 2 (4.2) | 1 (2.3) | 1 (2.4) | 1 (0.8) | 0 (0.0) | 5 (1.6) | |
| Unimportant (2–3 of Likert scale) | 14 (29.2) | 6 (13.6) | 5 (11.9) | 13 (10.9) | 9 (17.6) | 47 (15.5) | |
| Neutral (4–5 of Likert scale) | 19 (39.6) | 19 (43.2) | 15 (35.7) | 54 (45.4) | 24 (47.1) | 131 (43.1) | |
| Important (6–7 of Likert scale) | 11 (22.9) | 13 (29.5) | 16 (38.1) | 46 (38.7) | 14 (27.5) | 100 (32.9) | |
| Very important (8+ of Likert scale) | 2 (4.2) | 5 (11.4) | 5 (11.9) | 5 (4.2) | 4 (7.8) | 21 (6.9) | |
| TOTAL | 48 (100.0) | 44 (100.0) | 42 (100.0) | 119 (100.0) | 51 (100.0) | 304 (100.0) | |
| **HCP personal motivations to support people to self-care before the pandemic** | | | | | | | 0.007 |
| Empower the client/user | 40 (15.9) | 36 (22.0) | 39 (26.9) | 82 (15.4) | 43 (22.4) | 240 (18.7) | |
| It is more convenient to the client/user | 20 (7.9) | 10 (6.1) | 8 (5.5) | 40 (7.5) | 17 (8.9) | 95 (7.4) | |
| The option to self-care for some conditions is the superior mode of care | 34 (13.5) | 20 (12.2) | 20 (13.8) | 54 (10.2) | 18 (9.4) | 146 (11.4) | |
| Help prevent certain conditions or minimise likelihood of exacerbations | 40 (15.9) | 32 (19.5) | 28 (19.3) | 95 (17.9) | 31 (16.1) | 226 (17.6) | |
| To help me reduce my workload | 21 (8.3) | 3 (1.8) | 4 (2.8) | 11 (2.1) | 5 (2.6) | 44 (3.4) | |
| To help reduce pressure on scarce NHS resources | 32 (12.7) | 18 (11.0) | 17 (11.7) | 83 (15.6) | 22 (11.5) | 172 (13.4) | |
| To promote health literacy | 31 (12.3) | 23 (14.0) | 14 (9.7) | 65 (12.2) | 25 (13.0) | 158 (12.3) | |
| To promote the rational use of products & services | 32 (12.7) | 17 (10.4) | 12 (8.3) | 95 (17.9) | 24 (12.5) | 180 (14.0) | |
| Other (please specify) | 2 (0.2) | 5 (3.0) | 3 (2.1) | 7 (1.3) | 7 (3.6) | 24 (1.9) | |
| TOTAL | 252 (100.0) | 164 (100.0) | 145 (100.0) | 532 (100.0) | 192 (100.0) | 1285 (100.0) | |
| **Which aspects of self-care did you actively encourage prior to the pandemic?** | | | | | | | <0.001 |
| Building self-care capacity as part of routine professional care | 10 (3.5) | 14 (7.7) | 14 (6.6) | 41 (5.9) | 17 (6.2) | 96 (6.2) | |
| Dispensing & encouraging appropriate use of medicines | 15 (5.3) | 6 (3.3) | 0 (0.0) | 51 (7.3) | 5 (1.8) | 77 (4.9) | |
| Improving general wellbeing | 40 (14.1) | 32 (17.5) | 32 (15.2) | 95 (13.6) | 45 (16.3) | 244 (16.7) | |
| Improving mental health | 25 (8.8) | 13 (7.1) | 23 (10.9) | 41 (5.9) | 23 (8.3) | 125 (8.0) | |
| Making healthy lifestyle choices | 41 (14.5) | 37 (20.2) | 29 (13.7) | 105 (15.1) | 41 (14.9) | 253 (16.2) | |
| Managing cancer care | 4 (1.4) | 0 (0.0) | 0 (0.0) | 7 (1.0) | 0 (0.0) | 11 (0.7) | |
| Managing chronic long-term conditions | 16 (5.7) | 11 (6.0) | 11 (5.2) | 38 (5.5) | 18 (6.5) | 94 (6.0) | |
| Managing existing mental illness | 15 (5.3) | 5 (2.7) | 11 (5.2) | 30 (4.3) | 12 (4.3) | 73 (4.7) | |
| Managing Infectious diseases | 9 (3.2) | 1 (0.5) | 4 (1.9) | 19 (2.7) | 0 (0.0) | 33 (2.1) | |
| Managing minor illness & common conditions | 32(11.3) | 25 (13.7) | 14 (6.6) | 92 (13.2) | 26 (9.4) | 189 (12.1) | |
| Managing prescribing | 15 (5.4) | 6 (3.3) | 0 (0.0) | 51 (7.3) | 5 (1.8) | 77 (4.9) | |
| None of the above | 0 (0.0) | 0 (0.0) | 0 (0.0) | 3 (0.4) | 0 (0.0) | 3 (0.2) | |
| Other (please specify) | 3 (1.1) | 1 (0.5) | 1 (0.5) | 2 (0.3) | 3 (1.1) | 10 (0.6) | |
| Preventing infectious diseases | 18 (6.4) | 10 (5.5) | 9 (4.3) | 47 (6.8) | 10 (3.6) | 94 (6.0) | |
| Preventing mental illness | 21 (7.4) | 10 (5.5) | 11 (5.2) | 24 (3.4) | 13 (4.7) | 79 (5.1) | |
| Preventing non-communicable diseases | 19 (6.7) | 12 (6.6) | 6 (2.8) | 50 (7.2) | 14 (5.1) | 101 (6.5) | |
| **Total** | 283 (100.0) | 183 (100.0) | 211 (100.0) | 696 (100.0) | 276 (100.0) | 1559 (100.0) | |

*(Continued)*

**Table 2.** (Continued)

| | Doctor | Nurse | Service | Pharmacy | SP/Other | Total | p-value |
|---|---|---|---|---|---|---|---|
| | N (%) | N (%) | N (%) | N (%) | N (%) | N (%) | |
| **Perceived importance of self-care to patients since the advent of COVID-19 pandemic** | | | | | | | 0.27 |
| Very unimportant | 2 (4.2) | 0 (0.0) | 0 (0.0) | 2 (1.7) | 2 (3.9) | 6 (2) | |
| Unimportant | 0 (0.0) | 0 (0.0) | 0 (0.0) | 1 (0.8) | 0 (0.0) | 1 (0.3) | |
| Neutral | 2 (4.2) | 1 (2.3) | 2 (4.8) | 3 (2.5) | 0 (0.0) | 8 (2.6) | |
| Important | 1 (2.1) | 2 (4.6) | 2 (4.8) | 10 (8.4) | 9 (17.7) | 24 (7.9) | |
| Very Important | 42 (87.5) | 41 (93.2) | 38 (90.5) | 103 (86.6) | 40 (78.4) | 264 (86.8) | |
| Unknown | 1 (2.1) | 0 (0.0) | 0 (0.0) | 0 (0.0) | 0 (0.0) | 1 (0.3) | |
| **Perceived capacity/empowerment of service users to self-care since the advent of the pandemic** | | | | | | | 0.008 |
| Very unimportant (0–1 of Likert scale) | 3 (6.3) | 0 (0.0) | 1 (2.4) | 1 (0.8) | 0 (0.0) | 5 (1.6) | |
| Unimportant (2–3 of Likert scale) | 9 (18.8) | 3 (6.8) | 6 (14.3) | 11 (9.2) | 7 (13.7) | 36 (11.8) | |
| Neutral (4–5 of Likert scale) | 8 (16.7) | 16 (36.4) | 15 (35.7) | 19 (16.0) | 14 (27.5) | 72 (23.7) | |
| Important (6–7 of Likert scale) | 16 (33.3) | 17 (38.6) | 14 (33.3) | 44 (37.0) | 21 (41.2) | 112 (36.8) | |
| Very important (8+ of Likert scale) | 12 (25.0) | 8 (18.2) | 6 (14.3) | 44 (37.0) | 9 (17.6) | 79 (26.0) | |
| TOTAL | 48 (100.0) | 44 (100.0) | 42 (100.0) | 119 (100.0) | 51 (100.0) | 304 (100.0) | |
| **Main HCP motivations to support people to self-care since the advent of COVID-19 pandemic** | | | | | | | 0.12 |
| It can help reduce pressure on scarce NHS resources | 36 (15.4) | 28 (13.8) | 25 (14.5) | 102 (16.9) | 30 (14.7) | 221 (15.6) | |
| It is empowering to the client/user | 33 (14.1) | 40 (19.7) | 38 (22.1) | 92 (15.3) | 45 (22.1) | 248 (17.5) | |
| It is more convenient to the client/user | 19 (8.1) | 17 (8.4) | 15 (8.7) | 60 (10.0) | 19 (9.3) | 130 (9.2) | |
| The option to self-care for some conditions is the superior mode of care | 32 (13.7) | 25 (12.3) | 22 (12.8) | 67 (11.1) | 20 (9.8) | 166 (11.7) | |
| Help prevent certain conditions or minimise likelihood of exacerbations | 33 (14.1) | 34 (16.7) | 28 (16.3) | 94 (15.6) | 32 (15.7) | 221 (15.6) | |
| To help me reduce my workload | 25 (10.7) | 9 (4.4) | 6 (3.5) | 21 (3.5) | 5 (2.5) | 66 (4.7) | |
| To promote of health literacy | 23 (9.8) | 25 (12.3) | 20 (11.6) | 69 (11.4) | 27 (13.2) | 164 (11.6) | |
| To promote of the rational use of products & services | 32 (13.7) | 23 (11.3) | 16 (9.3) | 94 (15.6) | 20 (9.8) | 185 (13.1) | |
| Other (please specify) | 1 (0.4) | 2 (1.0) | 2 (1.2) | 4 (0.7) | 6 (2.9) | 15 (1.1) | |
| TOTAL | 234 (100.0) | 203 (100.0) | 172 (100.0) | 603 (100.0) | 204 (100.0) | 1416 (100.0) | |
| **Which Aspects of self-care did you actively encourage since the advent of the pandemic?** | | | | | | | <0.001 |
| Building self-care capacity as part of routine professional care | 32 (7.8) | 34 (9.7) | 25 (10.6) | 75 (7.4) | 33 (10.6) | 199 (8.6) | |
| dispensing & encouraging appropriate use of medicines | 27 (6.6) | 23 (6.6) | 4 (1.7) | 86 (8.5) | 8 (2.6) | 148 (6.4) | |
| Improving general wellbeing | 36 (8.7) | 39 (11.1) | 34 (14.5) | 94 (9.3) | 41 (13.2) | 244 (10.5) | |
| Improving mental health | 28 (6.8) | 31 (8.9) | 30 (12.8) | 67 (6.7) | 34 (11.0) | 190 (8.2) | |
| Making healthy lifestyle choices | 42 (10.2) | 38 (10.9) | 35 (14.9) | 95 (9.4) | 36 (11.6) | 246 (10.6) | |
| Managing cancer care | 18 (4.4) | 6 (1.7) | 1 (0.4) | 20 (2.0) | 5 (1.6) | 50 (2.2) | |
| Managing chronic long-term conditions | 34 (8.3) | 27 (7.7) | 20 (8.5) | 85 (8.4) | 26 (8.4) | 192 (8.3) | |
| Managing existing mental illness | 34 (8.3) | 27 (7.7) | 24 (10.2) | 66 (6.6) | 30 (9.7) | 181 (7.8) | |
| Managing Infectious diseases | 27 (6.6) | 12 (3.4) | 8 (3.4) | 49 (4.9) | 10 (3.2) | 106 (4.6) | |
| Managing minor illness & common conditions | 31 (7.5) | 24 (6.9) | 12 (5.1) | 93 (9.2) | 21 (6.8) | 181 (7.8) | |
| Managing prescribing | 27 (6.6) | 23 (6.6) | 4 (1.7) | 86 (8.5) | 8 (2.6) | 148 (6.4) | |
| None of the above | 0 (0.0) | 0 (0.0) | 0 (0.0) | 6 (0.6) | 1 (0.3) | 7 (0.3) | |
| Other (please specify) | 2 (0.5) | 2 (0.6) | 0 (0.0) | 4 (0.4) | 5 (1.6) | 13 (0.6) | |
| Preventing infectious diseases | 24 (5.8) | 23 (6.6) | 14 (6.0) | 64 (6.4) | 18 (5.8) | 143 (6.2) | |
| Preventing mental illness | 24 (5.8) | 21 (6.0) | 17 (7.2) | 51 (5.1) | 20 (6.5) | 133 (5.7) | |
| Preventing non-communicable diseases | 26 (6.3) | 20 (5.7) | 7 (3.0) | 66 (6.6) | 14 (4.5) | 133 (5.7) | |
| TOTAL | 412 (100.0) | 350 (100.0) | 235 (100.0) | 1007 (100.0) | 310 (100.0) | 2314 (100.0) | |

(Continued)

**Table 2.** (Continued)

| | Doctor | Nurse | Service | Pharmacy | SP/Other | Total | p-value |
|---|---|---|---|---|---|---|---|
| | N (%) | N (%) | N (%) | N (%) | N (%) | N (%) | |
| **Are you more or less likely to recommend & support your service users to self-care in your daily work as a result of the pandemic?** | | | | | | | 0.22 |
| **Significantly less likely** | 2 (4.2) | 0 (0) | 0 (0) | 2 (1.7) | 0 (0) | 4 (1.3) | |
| **Slightly less likely** | 1 (2.1) | 0 (0) | 0 (0) | 2 (1.7) | 1 (2.0) | 4 (1.3) | |
| **No change** | 3 (6.3) | 3 (6.8) | 6 (14.3) | 17 (14.3) | 9 (17.7) | 38 (12.5) | |
| **Slightly more likely** | 23 (47.9) | 12 (27.3) | 11 (26.2) | 38 (31.9) | 15 (29.4) | 99 (32.6) | |
| **Significantly more likely** | 18 (37.5) | 29 (65.9) | 25 (59.5) | 60 (50.4) | 26 (51.0) | 158 (52) | |
| **TOTAL** | 47 (100.0) | 44 (100.0) | 42 (100.0) | 119 (100.0) | 51 (100.0) | 303 (100.0) | |
| **Which aspect of self-care did you personally pursue during the pandemic?** | | | | | | | <0.001 |
| Building self-care capacity as part of routine professional care | 9 (4.9) | 18 (9.9) | 13 (8.2) | 41 (7.4) | 23 (10.3) | 103 (7.9) | |
| dispensing & encouraging appropriate use of medicines | 4 (2.2) | 6 (3.3) | 1 (0.6) | 32 (5.8) | 6 (2.7) | 49 (3.8) | |
| Improving general wellbeing | 31 (17.0) | 35 (19.2) | 28 (17.7) | 76 (13.7) | 33 (14.8) | 203 (15.6) | |
| Improving mental health | 20 (11.0) | 17 (9.3) | 19 (12.0) | 51 (9.2) | 25 (11.2) | 132 (10.1) | |
| Making healthy lifestyle choices | 33 (18.1) | 34 (16.7) | 30 (19.0) | 89 (16.1) | 38 (17.0) | 224 (17.3) | |
| Managing cancer care | 2 (1.1) | 0 (0.0) | 0 (0.0) | 6 (1.1) | 2 (0.9) | 10 (0.8) | |
| Managing chronic long-term conditions | 7 (3.8) | 7 (3.8) | 13 (8.2) | 17 (3.1) | 14 (6.3) | 58 (4.5) | |
| Managing existing mental illness | 6 (3.3) | 1 (0.5) | 10 (6.3) | 28 (5.1) | 11 (4.9) | 56 (4.3) | |
| Managing Infectious diseases | 6 (3.3) | 2 (1.1) | 4 (2.5) | 19 (3.4) | 4 (1.8) | 35 (2.7) | |
| Managing minor illness & common conditions | 14 (7.7) | 13 (7.1) | 8 (5.1) | 53 (9.6) | 14 (6.3) | 102 (7.9) | |
| Managing prescribing | 4 (2.2) | 6 (3.3) | 1 (0.6) | 32 (5.8) | 6 (2.7) | 49 (3.8) | |
| None of the above | 4 (2.2) | 3 (1.6) | 2 (1.3) | 5 (0.9) | 2 (0.8) | 16 (1.2) | |
| Other (please specify) | 2 (1.1) | 4 (2.2) | 1 (0.6) | 2 (0.4) | 4 (1.8) | 13 (1.0) | |
| Preventing infectious diseases | 19 (10.4) | 15 (8.2) | 16 (10.1) | 46 (8.3) | 19 (8.5) | 115 (8.9) | |
| Preventing mental illness | 12 (6.6) | 14 (7.7) | 8 (5.1) | 27 (4.9) | 15 (6.7) | 76 (5.9) | |
| Preventing non-communicable diseases | 9 (4.9) | 7 (3.8) | 5 (3.2) | 29 (5.3) | 7 (3.1) | 57 (4.4) | |
| **TOTAL** | 182 (100.0) | 182 (100.0) | 158 (100.0) | 553 (100.0) | 223 (100.0) | 1236 (100.0) | |
| **Perceived general barriers to self-care** | | | | | | | 0.98 |
| Barriers caused by new digital or telephone interactions | 20 (4.0) | 28 (5.6) | 24 (6.0) | 65 (4.8) | 29 (5.5) | 166 (5.1) | |
| Communication barriers due to hearing difficulties | 9 (1.8) | 19 (3.8) | 10 (2.5) | 51 (3.8) | 17 (3.2) | 106 (3.2) | |
| Dependency on professional role | 36 (7.1) | 23 (4.6) | 19 (4.7) | 66 (4.9) | 28 (5.3) | 172 (5.2) | |
| Difficulties in access to services & professionals | 27 (5.3) | 29 (5.8) | 28 (7.0) | 81 (6.0) | 27 (5.1) | 192 (5.9) | |
| Digital exclusion | 29 (5.7) | 35 (7.0) | 29 (7.2) | 83 (6.2) | 41 (7.8) | 217 (6.6) | |
| Health inequalities | 29 (5.7) | 35 (7.0) | 21 (5.2) | 74 (5.5) | 28 (5.3) | 187 (5.7) | |
| Health literacy | 26 (5.1) | 26 (5.2) | 15 (3.7) | 71 (5.3) | 26 (5.0) | 164 (5.0) | |
| Individual reluctance to engage or take responsibility | 37 (7.3) | 33 (6.6) | 29 (7.2) | 95 (7.1) | 32 (6.1) | 226 (6.9) | |
| Insufficient IT skills | 31 (6.1) | 34 (6.8) | 27 (6.7) | 81 (6.0) | 40 (7.6) | 213 (6.5) | |
| Interprofessional communication | 13 (2.6) | 17 (3.4) | 12 (3.0) | 54 (4.0) | 20 (3.8) | 116 (3.5) | |
| Lack of appropriate information to share with people | 20 (4.0) | 15 (3.0) | 15 (3.7) | 38 (2.8) | 9 (1.7) | 97 (3.0) | |
| Lack of consistency in approach or messages amongst professionals | 27 (5.3) | 25 (5.0) | 16 (4.0) | 81 (6.0) | 27 (5.1) | 176 (5.4) | |
| Lack of evidence of effectiveness of self-care interventions | 8 (1.6) | 7 (1.4) | 6 (1.5) | 17 (1.3) | 9 (1.7) | 47 (1.4) | |
| Lack of funding | 20 (4.0) | 12 (2.4) | 14 (3.5) | 51 (3.8) | 22 (4.2) | 119 (3.6) | |
| Lack of IT training | 12 (2.4) | 22 (4.4) | 6 (1.5) | 35 (2.6) | 15 (2.9) | 90 (2.7) | |
| Language barriers | 21 (4.2) | 22 (4.4) | 12 (3.0) | 47 (3.5) | 19 (3.6) | 121 (3.7) | |
| Other (please specify) | 8 (1.6) | 2 (0.4) | 3 (0.7) | 7 (0.5) | 4 (0.8) | 24 (0.7) | |
| Patient/client understanding of self-care | 36 (7.1) | 29 (5.8) | 27 (6.7) | 95 (7.1) | 31 (5.9) | 218 (6.7) | |
| Professional resistance to sharing responsibility | 12 (2.4) | 12 (2.4) | 11 (2.7) | 29 (2.2) | 11 (2.1) | 75 (2.3) | |

*(Continued)*

**Table 2.** (Continued)

| | Doctor | Nurse | Service | Pharmacy | SP/Other | Total | p-value |
|---|---|---|---|---|---|---|---|
| | N (%) | N (%) | N (%) | N (%) | N (%) | N (%) | |
| **Professional understanding of alternatives to face-to-face interactions** | 16 (3.2) | 13 (2.6) | 12 (3.0) | 36 (2.7) | 16 (3.0) | 93 (2.8) | |
| **Professional understanding of self-care** | 12 (2.4) | 7 (1.4) | 13 (3.2) | 26 (1.9) | 11 (2.1) | 69 (2.1) | |
| **Time constraints** | 35 (6.9) | 28 (5.6) | 23 (5.7) | 81 (6.0) | 31 (5.9) | 198 (6.0) | |
| **Transition of services from face-to-face to other formats** | 22 (4.3) | 30 (6.0) | 29 (7.2) | 78 (5.8) | 32 (6.1) | 191 (5.8) | |
| **TOTAL** | 506 (100.0) | 503 (100.0) | 401 (100.0) | 1342 (100.0) | 525 (100.0) | 3277 (100.0) | |

'improving general wellbeing' were consistently the two most frequently chosen options for promoting self-care in service users before and during the pandemic. Since the advent of the pandemic, recommending self-care to help 'manage existing mental illness' was the aspect of self-care that increased by the greatest margin followed by 'managing infectious disease' and 'building self-care capacity as part of routine professional care'. Overall, 84.6% of respondents were more likely to recommend self-care during the pandemic compared to before the pandemic.

**Barriers to self-care during the pandemic.** Across all professional groups, of the 23 potential barriers provided, the three most frequently chosen barriers were related to the service users as follows: 'individual reluctance to engage with health services' (74.6%), 'poor understanding of how to selfcare' (71.9%), and 'digital exclusion' (71.6%).

**Self-care resources and competencies needed to support self-care during the pandemic.** Three quarters (75.6%) of respondents considered that they personally had the resources and proficiency to promote self-care in their practice, but only 25.6% perceived their service users as adequately prepared to self-care. Only 28.5% agreed their service users had the competencies and resources to self-care (Table 2).

**Change in use of technology to support self-care during the pandemic.** All professional groups reported a significant increase in the use of the technology options listed during the COVID-19 pandemic. Overall, the three greatest increases in technology use were signposting to online self-care resources (86.1%), telephone use (78.9%) and the use of smartphone apps (77.1%) (S1 Table). The majority (86.0%) of HCPs reported that they were more likely to signpost service users to online self-care resources during the pandemic.

**Self-care practice after the pandemic.** Across all professional groups, 96.7% anticipated that they will continue to actively signpost service users to freely available self-care resources

**Table 3. Perceived importance of self-care to patients/service users before and after COVID-19.**

| | | Perceived importance of self-care after COVID-19 | | | | | | |
|---|---|---|---|---|---|---|---|---|
| | | Very unimportant | Unimportant | Neutral | Important | Very important | Total | p-value <0.001* |
| **Perceived importance of self-care before COVID-19** | **Very unimportant** | 3 | 0 | 2 | 2 | 15 | 22 | |
| | **Unimportant** | 0 | 0 | 0 | 1 | 0 | 1 | |
| | **Neutral** | 1 | 0 | 0 | 4 | 15 | 20 | |
| | **Important** | 2 | 0 | 4 | 13 | 77 | 96 | |
| | **Very important** | 0 | 1 | 2 | 4 | 157 | 164 | |
| **Total** | | 6 | 1 | 8 | 24 | 264 | 303 | |

* p-value obtained from McNemar's test

even after the pandemic abates. The majority (87.2%) reported that they will continue to promote more reliance on self-care practices amongst their service users, whereas 75.0% of HCPs felt they were personally more likely to practice self-care. Nearly three quarters (71.6%) of respondents considered that the advent of the pandemic had made the 'absolute case for self-care'. More than half (54.3%) viewed service users as being more likely to adhere to self-care practices as a first option and 58.1% felt that their service users are now better equipped or empowered to self-care since the advent of the pandemic (Table 2).

### Personal interviews

Nine respondents (4 male, 5 female) consented to an in-depth personal interview. The sample comprised of 5 doctors, 1 nurse, 1 ward sister and 2 pharmacists. Interviews were conducted online and lasted between 25–45 minutes. Four overarching themes were generated from the thematic analysis (S2 Table and S2 File) as follows: (1) professional approaches to self-care, (2) perceived patient/client attitudes to self-care, (3) issues arising since the advent of COVID, and (4) future reliance on self-care following the pandemic.

**Professional approaches to self-care.** In general, interviewees had a broad view of self-care in covering patient behaviours, in the home and community setting, pre-consultation and post-consultation with HCP. Most interview respondents considered self-care as a fundamental part of healthy living but recognised that pre-pandemic models of care and doctor-patient interactions did not always encourage personal empowerment and individual agency. Respondents expressed a fundamental need for a shared view of self-care that straddled professional and public realms, supported by changes in medical training. One respondent highlighted that in addition to supporting HCPs with training to deliver more person-centred care, patients and service users need to also be able to access quality assured information and resources to help build individual self-care capacity and capability.

> "...medical training and professional attitudes tended to place the patient as a helpless victim that needs rescuing by a professional or the NHS. It's really important we change this and empower patients to self-care where suitable.

**Perceived patient/client attitudes to self-care.** Respondents felt that limited access to routine services during the national lockdowns and prolonged periods of self-isolation meant that patients had to accept self-care as the mainstay of treatment. There was general agreement that certain ethnic groups, and in particular the elderly and those considered as clinically extremely vulnerable, experienced difficulties in access and changes in service delivery. These issues were thought to result from an acceptance of a paternalistic view of the patient's relationship with professionals, reflected in their need to ultimately obtain a doctor's opinion, even for self-manageable illnesses, including COVID.

> "...[with] open access to a trusted advisor there is no rational need to self-care. For some people, it's a case of why research things when you can ring the doctor and the lights in the hospital are on 24 hours a day?

Some respondents recognised that because there is not a 'one-size-fits-all' approach to promoting self-care in different patient groups, any lifestyle medicine and self-care recommendations need to be supported by easily accessible and quality assured self-care resources. The HCP interaction with the service user and attempts to signpost may otherwise appear to the patient as a form of 'abandonment' as opposed to 'empowerment'.

*". . . it's essential we give people the confidence to self-care along with necessary support otherwise there is a danger that people will feel abandoned rather than empowered".*

Most respondents felt that the NHS continues to promote itself as a "rescue service for victims" rather than supporting individuals in developing their own power and agency. One respondent suggested that NHS 111 and the use of decision support tools including online symptom checkers could help raise awareness about symptoms red flag conditions that need HCP support or emergent care compared to symptoms commonly experienced in self-limiting conditions. It was acknowledged that the widescale use of digital therapeutics and decision support tools may divert patients from seeking an unnecessary appointment with HCP for non-serious or self-limiting conditions, thus helping to reduce pressure on scarce NHS resources but also that any such reliance on online tools may exacerbate existing digital inequalities, especially with marginalised groups and already at-risk populations.

**Issues arising since the advent of COVID.** Respondents observed that all government guidance and most of the advice given by NHS 111 centred around getting people to self-care for coronavirus, either by avoiding an infection (prevention) or self-management of COVID-19 symptoms in the home setting. Advice and official guidance since the first national lockdown highlighted personal agency and risk avoidance to help "flatten the curve". This guidance was contemporaneous with various NHS England (NHSE)-led initiatives that saw the widespread acceptance and use of the NHS App, self-testing using rapid antigen test (RAT) kits, voluntary polymerase chain reaction (PCR) testing for those who tested positive on RAT, and the national launch of the @Home programme. At the same time, self-management of long-term conditions with point-of-care and diagnostic testing at home was quickly becoming the norm.

*"During the pandemic, the public moved to becoming totally self-caring by default. People bought their own thermometers, pulse oximeters, blood pressure monitors and personal hygiene products. Simple everyday things like alcohol wipes and hand sanitiser are now part of the mainstay of self-care. Lateral flow tests were a huge success, and their use was often supported with advice from pharmacists. People quickly learnt how to self-test, but this was not routine practice prior to the pandemic. It's just amazing to see how people's attitudes and reliance on self-care increased during the pandemic. I mean, you can argue that most people are now experts at self-care. Who would've thought that nearly everyone would know how to do an antigen test to self-diagnose when this was considered largely the preserve of healthcare professionals before the pandemic."*

A note of caution was expressed in some interviews with evidence of problems being exacerbated rather than being actively dealt with through self-care.

*". . . The fear that some people experienced since the advent of COVID-19 meant that there was suddenly a massive demand from people who've been by default self-caring and storing problems rather than dealing with them".*

Respondents felt that some groups, notably the elderly, did not- or could not- call or follow-up when appropriate for 'not wanting to bother' doctors and other professionals. A recurrent case study was how the number of cancer referrals had dropped significantly during the pandemic, with most referrals presenting with much later stage disease requiring intensive treatment and with ultimately poorer prognosis.

*"Some people felt empowered, but some cancers may well have been missed. . . presenting later with more extensive disease because of a perception that GP surgeries are closed."*

Technology played an important part in accessing healthcare remotely. Modalities including online consultations helped streamline access to HCPs and access to reliable online self-care resources during the pandemic. However, there was concern that the shift to telephone and online delivery, including eConsult may have exacerbated extant inequalities, especially in the elderly or segments of society that are digitally excluded.

*"[We] turned increasingly to mobile phone use although some older people did not have access to technology and do not understand it."*

. . . *"at times I felt that consultations with people you haven't engaged with before has been a process of alienation between the clinician and patient."*

Overall, respondents were sceptical that the benefits of increasing awareness about the importance of self-care would be permanent as some patients may perceive prolonged reliance on self-care as a kind of rationing, whereas other HCPs felt that it was only a matter of time before everyone naturally reverts to the traditional model of care once COVID was firmly in the rear-view mirror.

*"[some surgeries say] 'don't come here—go home' leaving patients to deal with their dependence on general practice which was not of their own making."*

*"There is a danger that encouraging self-care can be seen as rationing with a potential backlash."*

In reference to this, two interviewees referred to NHSE recommendation that primary care activity should revert to pre-COVID numbers of face-to-face consultations, even though NHSE did not show the evidence that this could result in better outcomes. It was felt that the numbers of consultations were considered to be more important than the quality.

*"The emphasis has been 'excess' not access."*

**Future reliance on self-care.** Despite the tragedies of COVID, all interview participants believed that self-care had become mainstream, with all advice promoting it. Rather than reverting to over-reliance on NHS staff, respondents felt the pandemic was an opportunity to embed some of the learning and the growing realisation that reliance on the self, and self-care, should be a mainstay of how some NHS services are delivered in the future:

*"It is an opportunity that needs to be grasped immediately. There has been a strong change in attitudes; self-care has been embraced by both sides".*

Respondents highlighted that the learning needs to be consolidated to resist central pressure to revert to a paternalistic system that have already proved difficult to maintain.

*"A fundamental change has been to give greater agency and autonomy to individuals to allow them to self-care whilst also providing a level of professional fulfilment by Zoom."*

*"It is not standard in routine general practice to discuss self-care options."*

Widespread access to good quality information, which is understandable, accessible, tailored, and inclusive, was required to ensure care remained optimal, with special attention for those who lack access to care the most.

*"There is an absolute need for information to be digestible and comprehensible, to make sense for the individual, and with specific advice on what to do next."*

The lack of self-care in the training of professionals, particularly the medical profession, needed to be addressed urgently to establish patient agency as the underpinning of care rather than a doctor centred view. This requires correcting in the postgraduate realm to a person-centred view of medicine. This was of equal importance in training for hospital specialties.

*"Self-care has been everybody's job but nobody's responsibility."*

*". . . a change in GP training is needed, otherwise the same attitudes to doctor centrality will continue."*

*"Self-care does not have its own domain or is even featured in many textbooks. . . and is not included in examinations."*

## Discussion

There have been few formal attempts to investigate attitudes to self-care of the general population [22–24] and fewer attempts to explore health and social care professionals' attitudes to self-care specifically [23, 25]. To our knowledge, this is the first study to explore HCP attitudes towards self-care since the advent of the COVID-19 pandemic. Specifically, this study sought to (1) Identify views of HCPs on self-care as a primary mode of care, (2) explore perceived barriers faced by service users to practice self-care, (3) identify the perceived barriers to self-care reported by healthcare workers, and (4) explore the changes in attitudes towards self-care following the advent of the COVID-19 pandemic.

We developed this study at a time of national lockdowns and when healthcare services were severely disrupted. We sought to understand HCP attitudes to self-care in the broadest sense, both for themselves and their perception of how important self-care was for their patients.

### Views of healthcare professionals on self-care as a primary mode of care

The importance of self-care as a meaningful, effective option was recognised in the UK as far back as 2002 [17, 23], particularly in the only dedicated NHS strategy on self-care. It has more recently been promoted internationally following the publication of the WHO guideline on self-care interventions in 2019 and 2022 [26]. The publication was opportune as it was contemporaneous with the advent of the COVID-19 pandemic that ushered in an era in which self-care became the norm by default for both people and professionals in the UK and internationally. A review of public surveys on self-care in the UK during the pandemic undertaken in 2022 (Smith, 2023, JIPCM, In Press), including the reports referred to above, indicated rapid and positive changes in attitudes to self-care in the general population, though with declining interest following the pandemic [24].

Respondents recognised a broad view of the different contexts of self-care, from making healthy lifestyle choices and improving general wellbeing and mental health through to prevention of infectious disease and managing long-term conditions. The majority of HCPs surveyed reported that they were more likely to recommend self-care as a means of personal

empowerment and to help reduce pressure on scarce NHS resources, but rarely to reduce their own workload. This is consistent with the position of the WHO and NHS England, as both entities consider self-care as a key enabler of health systems and not as a substitute for clinical healthcare. In the UK, national programmes that seek to promote personal empowerment and self-care using a task-shifting approach include the Proactive Care agenda and the NHS @Home programme [27].

The findings of our study also highlighted how HCPs' attitudes to their own self-care have changed significantly, with a greater majority agreeing that they would be personally more likely to practice self-care after the pandemic. Both professionals and the public appear to have embraced self-care and have begun to understand its benefits and importance, and all professional groups surveyed viewed self-care as empowering to patients. But person-centred concerns were raised, particularly in relation to digital and cultural exclusion of the most vulnerable and therefore the sustainability of the widespread adoption of self-care in this new setting.

## Perceived barriers faced by service users to practice self-care

Data were collected during the Covid pandemic, when much of the self-care advice in the UK including the 'Hands. Face. Space' public information campaign was delivered through public messaging supported by online and telephone advice from NHS 111 [9]. For practitioners who staffed the NHS 111 call centres, most of the advice given from its online prompts for those answering the telephones provided were by default under the title 'self-care'.

Although HCPs reported they had the resources to support self-care during the pandemic, they generally did not feel that service users were adequately prepared to self-care or had the competencies and resources to do so. This is reflected in professional views of the barriers to self-care of which the first two were 'reluctance to engage' and 'poor understanding'. 'Making healthy lifestyle choices' and 'improving general wellbeing' were consistently recommended by HCPs and these aspects were of great importance in their own lives, but there was a call to support these aspects with the provision of more quality-assured, understandable self-care education resource.

The third most often identified barrier to self-care for users was digital exclusion, which many respondents reported was exacerbated by the increased use of technology during the pandemic reported by all groups. In one instance, this prompted a report from a patient that they did not know whether the advice of the UK government and HCPs to self-care during the pandemic was 'empowerment or abandonment'. Another key emergent theme from personal interviews with HCPs was the rapid insistence on self-care practices during the national lockdowns through remote means without monitoring the true impact of these recommendations, coupled with a loss of follow-up of patients. This was a particular concern with the most vulnerable, including the elderly and certain ethnic populations for whom increased difficulties with access would be likely to exacerbate already existing inequalities. The issue has yet to be effectively addressed.

## Perceived barriers faced by healthcare workers

Study findings present a consistent professional view that patients' and clients' understanding of self-care and a reluctance to engage with professionals on the matter is the single greatest barrier to self-care. This echoes the findings of a previous survey of public views on self-care [28] in 2005 by the Department of Health when 88% of people declared they were not encouraged to self-care when visiting a GP, with most declaring they were already practising it. In 2011, McAteer et al's results tended to support the public view, with only 8% seeking GP

opinion on a range of symptoms and 50% doing nothing [29]. This might explain some of the difference in understanding, with self-care already having been practiced by individuals and carers prior to seeking advice.

The necessarily rapid change in delivery of services during the pandemic also involved a paradigm shift from what was considered to be a state of dependency on professionals to one of supported but self-reliant self-care. Both people and professionals expressed concerns about this change.

The 2022 review cited above (Smith, 2023, In Press) also examined the responses of service providers to the loss of face-to-face contacts as reflected in the Self Care Forum Coronavirus Innovation Awards [7]. The rapid conversion to mainly online and telephone delivery instead of traditional face-to-face services for vulnerable featured strongly, but digital exclusion was a consistent major issue.

## Changes in attitudes towards self-care following the advent of the COVID-19 pandemic

Although professionals reported that they would continue to support patients/clients to self-care, they were sceptical that people would continue to self-care, potentially perceiving it after the pandemic as a form of rationing of access. Surveys of public views on self-care undertaken by the PAGB in 2020 [1], 2021 [3] and 2022 [24] suggested that this positive public attitude was receding and concerns were raised that beneficial changes might not survive a return to 'normality'.

International comparators are scarce. In 2020 GSK/IPSOS/EMEA conducted a European survey across four countries around the time of the 2020 PAGB Survey and published a summary online [2]. It has not been repeated, so the trends cannot be established, however at the majority of people considered it important to take their health into their own hands to relieve pressure on healthcare systems– 84% in Spain, 77% in the UK, 75% in Italy and 63% in Germany. This is a similar figure to our study in which motivation of professionals to promote self-care during the pandemic 'in order to reduce pressure on scarce NHS resources' rose to 72.7% from 56.6% pre-pandemic. Given a list of minor symptoms and asked whether they would see a doctor or go to A&E as a first port of call, 50% in Spain 40% and Germany and only 10% would do so in the UK. This is close to McAteer's findings in 2011, in which only 8% of respondents would seek a GP opinion when experiencing a range of symptoms [29].

Some specific aspects of self-care have been explored. For example, Moore and Coggins investigated attitudes to shared wound care across seven countries. On the issue of overall attitudes to patient involvement, results are not broken down by country but an average of 42% of professionals were either very or extremely positive about patient engagement. When asked to estimate the percentage of patients involved in their own wound care, results ranged from 77% in China to 41% in France, with the UK at 58% [30].

Mollica et al. [31] reviewed individual professionals' attitudes to their own self-care and in particular with relation to burnout caused by physical and mental exhaustion and in the USA, Miller and Cassar [32] investigated the impact of COVID-19 on the self-care practices of self-identified healthcare and social workers and found that participants who identified as married, financially stable, working non-remotely, and in good physical/mental health engaged in significantly more self-care practices than other participants.

The subtext in the UK analyses is the issue of the introduction and perceived limitations of self-care supported through remote contacts compared to traditional face-to-face consultations. The provision of telephone and online consultations was originally seen as a progressive step, fuelled by the popular success of remote GP services such as Babylon [33]. It was

encouraged in the NHS well before the onset of the pandemic. The NHS Long Term Plan, published in January 2019, committed to the right of every patient to be offered 'digital-first' primary care by 2023/24 [34, 35]. The pressure to achieve a digital future for the NHS was being heralded by the Secretary of State for Health as late as March 2021 [36] and enshrined in GP contract requirements in October 2021 [37], which insisted on the offer and promotion of online services.

Whilst the insistence on these online obligations has continued, demand for face-to-face has escalated, perhaps confirming professional concerns that problems may have been shelved rather than dealt with. Although professionals responded that they would continue to support patients and clients to self-care and agreed it had now become mainstream, they were sceptical that this would be encouraged.

This view has been supported by the publication of data on individual practices' appointments [38] and the subsequent praising of those that have fewer remote offerings. This has led to a situation in which the benefits of empowering remote supported self-care may be lost, the first point of call again being a professional. This is despite its benefits in promoting personal agency and control.

A previous study into the use of GP consultations in 2016 suggested that up to 18% of GP appointments (57 million appointments a year) are for conditions that could be self-treated [23]. At the time, around 90% of these consultations were ended with the provision of a prescription. The concern arises again as expressed by an interviewee that with the easy availability of professional appointments, attendance for reassurance instead of searching for self-care advice was logical.

There is a lack of concordance between the understanding of the public and professionals on self-care and there remains an ambivalence towards it in the UK public. This is reflected in political uncertainty on the subject, with self-care during a pandemic seeming to be accepted but with the expectation that there would be a return to the 'normality' of seeking professional advice when it was over.

This survey was undertaken towards the end of the acute stage of the pandemic and professionals were already voicing concerns about the reality of public readiness for and capacity to self-care. Further concerns were also raised about the likelihood of a retreat from self-care following the pandemic. Both concerns appear to have been confirmed. The concern reflected in this study, that any gains require reinforcement of the importance of self-care, including good information and changes in medical training to include self-care, have yet to be heeded. Political support for the concept of self-care is also required if the benefits of self-care to be realised.

## Study implications

The public has its own understanding of self-care, and this is reflected in the 5.4 billion items appearing in a Google search for 'self-care' (June 2023). However, many of these resources may include recommendations that are not evidence-based. Professionals may also access over 250,000 peer-reviewed papers in a PubMed search on self-care (June 2023), but it remains impossible for all HCPs to be expert at recommending self-care to others, or indeed being exemplary self-carers themselves. It is also true that routine interaction with HCPs is minimal, such that even for an older person with a long-term condition they may have face-to-face contact with a professional for 10 hours of a year, leaving 8750 hours of self-care which informs their approach. Technology also played an important part in accessing healthcare remotely, and this trend will likely continue given the emergence of the self-driven healthcare (SDH) movement [39].

These realisations coupled to our study findings have huge implications on how to promote awareness, knowledge and democratise access to self-care products, services and interventions. First, since HCPs' view self-care as empowering but good evidence-based information is required to support them, it is recommended to develop and maintain a centra repository of evidence-based self-care resources that could be accessed by HCPs and the public. The Self-Care Forum UK is in part trying to accomplish this vis-à-vis it's Self-Care Factsheets initiative.

Second, because the relationship with HCPs continues to be highly valued and reassuring to patients, and despite a significant shift to online and teleconsultations, the push to digital first, although important, should not be seen as a replacement for personal contacts.

Crucially, the general public and patients with self-limiting and chronic long-term diseases require further education to improve self-care capability across all pillars of self-care. One way to achieve this would be to use a structured microlearning approach [40] and embed self-care education in schools, in workplaces and in different settings throughout the life course. The medical and nursing curricula should also integrate elements relevant to promoting individual self-care capability, including modalities such as coaching, behaviour change interventions and greater emphasis on lifestyle medicine approaches.

At the macro level, it is crucial for individual countries to develop a national self-care strategy. In the UK, the National Self-Care Strategy first presented in 2005 [22] is now being revisited, and this may in the near future lead to a self-care blueprint that the UK government could recommend to health and social care services with the potential to impact individuals from across the age groups and from all walks of life.

Our study findings highlight an urgent need for a coherent and consistent national view on self-care. To achieve this there is a need for education of the public, professionals and the political world on the practicality and benefits of self-care if its short-term acceptance during the pandemic is to become permanent. Personal attitudes and understanding of self-care have undergone rapid and significant changes as result of the pandemic. Whether or not the changes in attitudes will persist will depend upon future response by the public, professionals, and politicians to support and encourage these changes.

## Limitations

To our knowledge, this is the first study to investigate HCP attitudes to self-care and how this may have changed following the national lockdowns to tackle the COVID-19 pandemic. We acknowledge that self-care and self-management are technically different but related concepts, whereas in some instances the terms are used interchangeably- even in the health and social care profession. Despite this, we sought to not focus too much on delineating self-care and self-management in this study to keep the focus centered around changing attitudes to self-care in its broadest sense.

A key limitation of the study was the relatively small sample size of 304 HCPs using an electronic survey. However, this allowed us to investigate key trends in changes in attitudes, and to consider how these differed when broken down into professional groups for all questions. Thus, whereas 41 respondents were grouped under 'services & physiotherapy professionals' and 49 respondents were grouped under social prescriber or other (i.e., carer, non-GP doctor, commissioner of health), this combined subsample only represents 90/304 (29%) of the total, and we acknowledge that only a fraction of these may in fact be 'social care'. This implies that our results were heavily skewed to pharmacists (n = 119) and to primary care staff (n = 47) and were not necessarily representative of the social care workforce which to date remains largely understudied.

Another key limitation was that because all responses were self-reported, it is possible that respondents may not have held the same views that are representative of their professional tier which makes the data less generalisable. We acknowledge that additional interviews may have resulted in the identification of other emergent themes, particularly when trying to character-ise differences in attitudes between different healthcare workforce groups (e.g., comparing the opinion of doctors to social prescribers). Despite this, the pragmatic sample size of the per-sonal interview component was deemed sufficient as saturation of themes was accomplished [41]. We appreciate that some selection bias was also possible such that only those with inter-net access and an interest in the topic participated in the eSurvey or agreed to be interviewed. As the total number of participants was only 304 participants, we also cannot assume their per-spectives are fully representative of the UK health or social care workforce. The findings and insights from this pilot could inform the development of a larger and more complex study, including a more diverse cross-section of HCPs including policymakers and commissioners of health and wellbeing support services.

Another limitation was not including a working definition of self-care in the introduction section of the eSurvey. Our decision to not include a definition was based on streamlining the way the survey is presented (e.g. small introduction with minimum text) and the assumption that the intended participants would have prior knowledge of self-care ad that encompassing self-management of minor ailments. Future studies could address this limitation by including a working definition of self-care in the introductory section of the survey.

## Conclusion

This is the first study to formally examine the attitudes and practices of healthcare profession-als' practices in relation to self-care since the advent of the COVID-19 pandemic, which we show has changed significantly over the last two years. The unified UK government response to tackle the pandemic has made the absolute case for self-care, but the cross-section of the healthcare workforce surveyed were uncertain if service users would adhere to self-care prac-tices as a first option when the pandemic abates, and NHS services resume with increased numbers of traditional face-to-face contacts. Further formal study is required to investigate if these signal changes in attitudes and practices will be sustained or enhanced, to further exam-ine public perceptions of self-care and how this will be articulated across the newly inaugu-rated integrated care systems and emergent UK self-care strategies.

## Supporting information

**S1 Table. Resources, competencies, technology and future.**
(PDF)

**S2 Table. CAPPS Thematic framework table.**
(DOCX)

**S1 File. Survey export.**
(PDF)

**S2 File. Interview coding of overarching themes.**
(PDF)

**S3 File. Interview guide.**
(PDF)

**S4 File. COREQ checklist.**
(DOCX)

**S5 File. CHERRIES checklist.**
(DOCX)

**S6 File. Rawdata.**
(XLSX)

## Acknowledgments

The authors thank the Self-Care Forum for disseminating the link to the survey.

## Author Contributions

**Conceptualization:** Peter Samuel Smith, Aos Alaa, Eva Riboli Sasco, Austen El-Osta.

**Formal analysis:** Peter Samuel Smith, Aos Alaa, Eva Riboli Sasco, Emmanouil Bagkeris.

**Funding acquisition:** Aos Alaa, Eva Riboli Sasco.

**Investigation:** Peter Samuel Smith.

**Methodology:** Aos Alaa.

**Visualization:** Peter Samuel Smith, Aos Alaa, Eva Riboli Sasco, Emmanouil Bagkeris, Austen El-Osta.

**Writing – original draft:** Peter Samuel Smith.

**Writing – review & editing:** Peter Samuel Smith, Aos Alaa, Eva Riboli Sasco, Austen El-Osta.

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
