## [Decision Letter · Decision Letter 0]

1 Jun 2023

PONE-D-23-05340How has COVID-19 changed health and social care professionals' attitudes to self-care? A mixed method research study.PLOS ONE

Dear Dr. Smith,

Thank you for submitting your manuscript to PLOS ONE. After careful consideration, we feel that it has merit but does not fully meet PLOS ONE’s publication criteria as it currently stands. Therefore, we invite you to submit a revised version of the manuscript that addresses the points raised during the review process.

We look forward to receiving your revised manuscript.

Kind regards,

Vijayaprakash Suppiah, PhD

Academic Editor

PLOS ONE

Journal Requirements:

Reviewers' comments:

Reviewer's Responses to Questions

**Comments to the Author**

1. Is the manuscript technically sound, and do the data support the conclusions?

Reviewer #1: Partly

Reviewer #2: Partly

2. Has the statistical analysis been performed appropriately and rigorously? 

Reviewer #1: I Don't Know

Reviewer #2: N/A

3. Have the authors made all data underlying the findings in their manuscript fully available?

Reviewer #1: Yes

Reviewer #2: Yes

4. Is the manuscript presented in an intelligible fashion and written in standard English?

Reviewer #1: Yes

Reviewer #2: Yes

5. Review Comments to the Author

Reviewer #1: Thank you for inviting me to review this paper. The paper is well written and clearly structured. Overall, the description of the conduct of the study is good. However, there are some areas which are unclear.

In the data analysis section, the authors say that Chi-square test was used to compare the responses to the survey by different groups but this does not appear to be presented in the results section. They are so state that ‘McNemar’s test was used to compare paired data from different time horizons’. As the paper reports on the results of a single survey, albeit one conducted over a six-month time period, I am unclear what the different time horizons refer to are. If this simply refers to perceptions of self-care before and during the pandemic, this should be made clear. Also, there is no mention of the results of this analysis in the results section.

With regard to the qualitative interviews, the Materials and Methods section states that a sub-sample of none survey respondents were ‘selected’ for the interview. However, under in the results section (p15) it appears that only nine survey respondents consented to be interviewed. If this is the case, it should be made clear that the interviewees were not selected but rather were a convenience sample of those who volunteered.

My biggest concern with this paper is the underlying study, and in particular the representativeness of the survey and the conclusions that can be drawn from it. I fully appreciate that eliciting responses to surveys during COVID-19 was difficult but the results are heavily skewed to one professional group i.e. pharmacists, and to primary care. I also feel it is stretching a point to suggest that the survey encompasses both health and social care professionals. Most people would understand ‘social care professionals’ to be care workers in residential or community setting, and social care staff working for social services departments. Groups such as social prescribers are essentially primary care staff. It would be more accurate to describe this as a survey of health care professionals. The authors do not report on which sectors of health care the non-primary care professionals worked in. This makes it hard to contextualise the responses, for example in relation to self-care for existing mental health problems.

I also have some concerns about what conclusions can be safely drawn from specific questions in the survey. For example, the choices in some questions seemed odd or inconsistent, or options which may have been important were absent. For example, in the barriers to self-care, poverty or lack of financial resources was not given as an option. My knowledge in this area is limited, but I wonder if there should have been more discussion of the difference between ‘self-care’ and ‘self-management’. Attitudes to and the skills needed for supporting patients will vary between say self-care of minor illnesses and the self-management of mental and physical multimorbidity.

Nevertheless, the paper does raise some interesting points about changing attitudes to self-care in its broadest sense and the changes needed in professional education to support this shift.

Reviewer #2: The study the authors conducted, which utilizes mixed methods, offers valuable insights into the changes in self-reported professional attitudes, perceptions, and self-care practices among health and social care professionals before and during the pandemic. While the study contributes significantly to our understanding of these changes, I have a few suggestions for the authors' consideration to improve the clarity and organization of this paper.

1. Clarify the Definition of "Self-Care": It is essential to provide a clear definition of "self-care" in your study. Besides including the definition in the Intro or Method section, this clarification should be included in the online survey and before the semi-structured interviews to ensure that participants have a shared understanding of the term. Since the meaning of self-care can vary depending on the context and individual backgrounds, a clear definition will help participants provide accurate responses and enable the identification of concrete interventions to improve self-care based on the study findings.

2. Specify the Focus of the Study: Throughout the manuscript, please clarify whether your study aimed to examine changes in self-care attitudes and practices in providers themselves or attitudes toward self-care when providers were taking care of their patients or teaching patients about the importance of self-care during practice. As the study encompasses both patient and provider self-care, it would be beneficial to organize the results based on these two categories. This clarification will enhance the reader's understanding of the study's objectives and outcomes.

3. Discuss Implications Based on Study Findings: In the Discussion section, please include a short paragraph suggesting potential implications based on the study findings. By doing so, the authors can provide practical insights for healthcare organizations, professionals, and policymakers. Consider discussing specific recommendations or interventions that could be implemented to address the barriers to self-care identified in this study. These implications will enhance the overall impact of this research and offer tangible suggestions for future practice.

4. Provide Information on Participant Selection for Interviews: Please include a flowchart or a brief description explaining how the authors selected specific individuals to participate in the semi-structured interviews. Clarify whether these individuals were representative of the survey population and discuss any inclusion or exclusion criteria used in the selection process. This information will provide transparency and strengthen the validity of the qualitative data collected.

6. PLOS authors have the option to publish the peer review history of their article (what does this mean?). If published, this will include your full peer review and any attached files.

Reviewer #1: No

Reviewer #2: No

---

## [Author Response · Author response to Decision Letter 0]

1 Jul 2023

Please find below our response to helpful comment from you and our reviewers. Comments are listed in full, with our responses in bold thereafter. 

Editors’ comments:

Comment 1: Please ensure that your manuscript meets PLOS ONE style requirements, including those for file naming. 

Response: we have thoroughly re-checked the manuscript and corrected as per PLOS templates provided.

Comment 2: We note that you have indicated that data from the study are available on request. PLOS ONE only allows data to be available upon request if there are legal or ethical restrictions on sharing data publicly.

Response: we are pleased to be able to inform you that there are no restrictions with availability of this data and the anonymized data set necessary to replicate the study is now included as supporting information in supplementary S6 File. RAW_DATA.

Reviewers‘ comments

Reviewer 1

Comment 1: In the data analysis section, the authors say that Chi-square test was used to compare the responses to the survey by different groups but this does not appear to be presented in the results section. They are so state that ‘McNemar’s test was used to compare paired data from different time horizons’. As the paper reports on the results of a single survey, albeit one conducted over a six-month time period, I am unclear what the different time horizons refer to are. If this simply refers to perceptions of self-care before and during the pandemic, this should be made clear. Also, there is no mention of the results of this analysis in the results section.

Response: We refer to perceptions of self-care before and during the pandemic and this has now been made clear in the results and discussion section “McNemar's test was used to compare paired data from different time horizons (i.e. before and during the pandemic) regarding HCPs’ perceptions of whether self-care was important to service users. This is also now mentioned in the results section “There was agreement that the importance of self-care has increased markedly during the pandemic, p<0.001 (Table 3)”. 

Comment 2: With regard to the qualitative interviews, the Materials and Methods section states that a sub-sample of nine survey respondents were ‘selected’ for the interview. However, under in the results section (p15) it appears that only nine survey respondents consented to be interviewed. If this is the case, it should be made clear that the interviewees were not selected but rather were a convenience sample of those who volunteered.

Response: Thank you for highlighting this. We updated the text in the manuscript to confirm that the interviews resulted from a convenience sample of respondents who volunteered and consented to be contacted.

Comment 3: My biggest concern with this paper is the underlying study, and in particular the representativeness of the survey and the conclusions that can be drawn from it. I fully appreciate that eliciting responses to surveys during COVID-19 was difficult but the results are heavily skewed to one professional group i.e. pharmacists, and to primary care. I also feel it is stretching a point to suggest that the survey encompasses both health and social care professionals. Most people would understand ‘social care professionals’ to be care workers in residential or community setting, and social care staff working for social services departments. Groups such as social prescribers are essentially primary care staff. It would be more accurate to describe this as a survey of health care professionals. The authors do not report on which sectors of health care the non-primary care professionals worked in. This makes it hard to contextualise the responses, for example in relation to self-care for existing mental health problems.

Response: Thank you for raising this important point. Although we intended to recruit a representative cross-section of health and social care professionals but did not recruit sufficient participants from social care, and we acknowledge also that we did not give the respondents an opportunity to specify their exact designation. Thus, whereas 41 respondents were grouped under ‘services & physiotherapy professionals’ and 49 respondents were grouped under Social prescriber or other (i.e., carer, non-GP doctor, commissioner of health), this total subsample only represents 90/304 (29%) of the total, and we acknowledge that only a fraction of these may in fact be ‘social care’. On this basis, and thanks to your recommendation, we included additional text to emphasises that our results are heavily skewed to pharmacists (n=119) and to primary care staff (n=47). This is a key limitation of the study which we have now highlighted in the limitations section (discussion). In view of this important observation, we adjusted the text to emphasise that this survey encompasses healthcare professionals (HCPs) but excluded mention of social care specifically. We also reflected this change in the study title. 

Comment 4: I also have some concerns about what conclusions can be safely drawn from specific questions in the survey. For example, the choices in some questions seemed odd or inconsistent, or options which may have been important were absent. For example, in the barriers to self-care, poverty or lack of financial resources was not given as an option.

My knowledge in this area is limited, but I wonder if there should have been more discussion of the difference between ‘self-care’ and ‘self-management’. Attitudes to and the skills needed for supporting patients will vary between say self-care of minor illnesses and the self-management of mental and physical multimorbidity. Nevertheless, the paper does raise some interesting points about changing attitudes to self-care in its broadest sense and the changes needed in professional education to support this shift.

Response: Thank you for raising these important points. Q18 (What do you feel are the main barriers to self-care?) included 24 options. Option 6 was ‘Lack of funding” & option 15 was “Lack of equipment/hardware with which to communicate”. Both of these options could be considered as proxy-measures to financial deprivation. We also included an “other” option (option 24) where respondents could include free-text in which no respondent included the two suggested barriers. 

We acknowledge that self-care and self-management are technically different but related concepts, whereas in some instances the terms are used interchangeably- even in the health and social care profession. The paper does include some key discussion points about both aspects, but we sought to not focus too much on delineating self-care and self-management in the paper (and survey) to keep the focus centered around changing attitudes to self-care in its broadest sense. We have included some paragraphs in the discussion section to highlight this as per your recommendation.

Reviewer 2

Comment 1: Clarify the Definition of "Self-Care": It is essential to provide a clear definition of "self-care" in your study. Besides including the definition in the Intro or Method section, this clarification should be included in the online survey and before the semi-structured interviews to ensure that participants have a shared understanding of the term. Since the meaning of self-care can vary depending on the context and individual backgrounds, a clear definition will help participants provide accurate responses and enable the identification of concrete interventions to improve self-care based on the study findings.

Response: Thank you for raising this important point. A paper by Godfrey in 2011 highlighted 113 extant definitions of self-care in the academic literature. Even the WHO has 5 definitions of self-care (the latest being proposed in 2019). 

Despite this, we acknowledge that it would have been very useful to include a working definition of self-care (both in the introduction of the eSurvery & in the manuscript). 

When designing the survey, we considered it may be helpful to engage with respondents about self-care in the way that they conceive it (considering that our sample was from allied/health/social care- we felt there may be a ‘common’ understanding that this also encompasses aspects relevant to self-management of minor, self-limiting illnesses. We were also concerned that including a definition would increase the introduction section of the survey & that this may detract some potentially eligible participants from completing the survey. In hindsight it would have been useful to include a working definition (see section below). We now highlight this omission as a limitation in the limitations section (discussion).

As for the readership of the manuscript, we have actioned your excellent recommendation and included the most widely used working definition of self-care by WHO as “the ability of individuals, families and communities to promote their own health, prevent disease, maintain health, and to cope with illness and disability with or without the support of a health worker”.

Comment 2: Specify the Focus of the Study: Throughout the manuscript, please clarify whether your study aimed to examine changes in self-care attitudes and practices in providers themselves or attitudes toward self-care when providers were taking care of their patients or teaching patients about the importance of self-care during practice. As the study encompasses both patient and provider self-care, it would be beneficial to organize the results based on these two categories. This clarification will enhance the reader's understanding of the study's objectives and outcomes.

Response: Thank you for raising this point. We developed this study at a time of national lockdowns and when health and care services were severely disrupted. We sought to understand HCP attitudes to self-care in the broadest sense (for themselves and their perception of how important self-care was for their patients). We therefore intended for the survey to focus on both aspects. We organised the results to present data regarding professional attitudes and approached to self-care, and subsequently presented the data relevant to perceiving patient/client attitudes to self-care. We have also included a short paragraph (the second paragraph in the discussion section) to highlight the study focus.

Comment 3: Discuss Implications Based on Study Findings: In the Discussion section, please include a short paragraph suggesting potential implications based on the study findings. By doing so, the authors can provide practical insights for healthcare organizations, professionals, and policymakers. Consider discussing specific recommendations or interventions that could be implemented to address the barriers to self-care identified in this study. These implications will enhance the overall impact of this research and offer tangible suggestions for future practice.

Response: Thank you for highlighting this. We have now included a new section prior to the study limitation section to discuss the implications of the study findings. This culminates in a series of recommendations for the consideration of policy makers to advance the vision for greater awareness about self-care for patient and public benefit.

Comment 4: Provide Information on Participant Selection for Interviews: Please include a flowchart or a brief description explaining how the authors selected specific individuals to participate in the semi-structured interviews. Clarify whether these individuals were representative of the survey population and discuss any inclusion or exclusion criteria used in the selection process. This information will provide transparency and strengthen the validity of the qualitative data collected.

Response: Thank you for raising this important point. Although we intended to recruit a representative cross-section of health and social care professionals but did not recruit sufficient participants from social care, and we acknowledge also that we did not give the respondents an opportunity to specify their exact designation. The interviewees were not selected but were recruited as a convenience sample from those who volunteered to partake in the study after meeting the inclusion criteria (over 18 years of age, English speaking, and currently working in nursing, health, pharmacy or social care setting). Participants who did not consent to be interviewed were excluded. We added the relevant text in the methods section to highlight these additions.

We would like to thank you and the reviews for your helpful and constructive criticism which is produced a far more robust and insightful study study.

---

## [Decision Letter · Decision Letter 1]

11 Jul 2023

How has COVID-19 changed healthcare professionals' attitudes to self-care? A mixed method research study.

PONE-D-23-05340R1

Dear Dr. Smith,

We’re pleased to inform you that your manuscript has been judged scientifically suitable for publication and will be formally accepted for publication once it meets all outstanding technical requirements.

Kind regards,

Vijayaprakash Suppiah, PhD

Academic Editor

PLOS ONE

Reviewers' comments:

Reviewer's Responses to Questions

**Comments to the Author**

1. If the authors have adequately addressed your comments raised in a previous round of review and you feel that this manuscript is now acceptable for publication, you may indicate that here to bypass the “Comments to the Author” section, enter your conflict of interest statement in the “Confidential to Editor” section, and submit your "Accept" recommendation.

Reviewer #1: All comments have been addressed

2. Is the manuscript technically sound, and do the data support the conclusions?

Reviewer #1: Yes

3. Has the statistical analysis been performed appropriately and rigorously? 

Reviewer #1: I Don't Know

4. Have the authors made all data underlying the findings in their manuscript fully available?

Reviewer #1: Yes

5. Is the manuscript presented in an intelligible fashion and written in standard English?

Reviewer #1: Yes

6. Review Comments to the Author

Reviewer #1: Thank you for addressing both reviewers comments so thoroughly. I particularly like the 'study implications' section you have added - this really strengthens the paper.

7. PLOS authors have the option to publish the peer review history of their article (what does this mean?). If published, this will include your full peer review and any attached files.

Reviewer #1: No

---

## [Editor Report · Acceptance letter]

14 Jul 2023

PONE-D-23-05340R1 

How has COVID-19 changed healthcare professionals' attitudes to self-care? A mixed methods research study. 

Dear Dr. Smith:

I'm pleased to inform you that your manuscript has been deemed suitable for publication in PLOS ONE. Congratulations! Your manuscript is now with our production department. 

Kind regards, 

on behalf of

Dr. Vijayaprakash Suppiah 

Academic Editor

PLOS ONE